# Silencing of Sb*PPCK1-3* Negatively Affects Development, Stress Responses and Productivity in Sorghum

**DOI:** 10.3390/plants12132426

**Published:** 2023-06-23

**Authors:** Jesús Pérez-López, Ana B. Feria, Jacinto Gandullo, Clara de la Osa, Irene Jiménez-Guerrero, Cristina Echevarría, José A. Monreal, Sofía García-Mauriño

**Affiliations:** 1Departamento de Biología Vegetal y Ecología, Facultad de Biología, Universidad de Sevilla, Avenida Reina Mercedes nº 6, 41012 Seville, Spain; jperezl@us.es (J.P.-L.); anabelen@us.es (A.B.F.); jacintogt@us.es (J.G.); cdelaosa@us.es (C.d.l.O.); echeva@us.es (C.E.); monreal@us.es (J.A.M.); 2Departamento de Microbiología, Facultad de Biología, Universidad de Sevilla, Avenida Reina Mercedes nº 6, 41012 Seville, Spain; ijimgue@us.es

**Keywords:** phospho*enol*pyruvate carboxylase, phospho*enol*pyruvate carboxylase kinase, protein phosphorylation, seed production, seed quality, *Sorghum bicolor*

## Abstract

Phosphoenolpyruvate carboxylase (PEPC) plays central roles in photosynthesis, respiration, amino acid synthesis, and seed development. PEPC is regulated by different post-translational modifications. Between them, the phosphorylation by PEPC-kinase (PEPCk) is widely documented. In this work, we simultaneously silenced the three sorghum genes encoding PEPCk (Sb*PPCK1*-*3*) by RNAi interference, obtaining 12 independent transgenic lines (*Ppck1-12* lines), showing different degrees of Sb*PPCK1-3* silencing. Among them, two T2 homozygous lines (*Ppck-2* and *Ppck-4*) were selected for further evaluation. Expression of Sb*PPCK1* was reduced by 65% and 83% in *Ppck-2* and *Ppck-4* illuminated leaves, respectively. Expression of Sb*PPCK2* was higher in roots and decreased by 50% in *Ppck-2* and *Ppck-4* in this tissue. Expression of Sb*PPCK3* was low and highly variable. Despite the incomplete gene silencing, it decreased the degree of phosphorylation of PEPC in illuminated leaves, P-deficient plants, and NaCl-treated plants. Both leaves and seeds of *Ppck* lines had altered metabolic profiles and a general decrease in amino acid content. In addition, *Ppck* lines showed delayed flowering, and 20% of *Ppck-4* plants did not produce flowers at all. The total amount of seeds was lowered by 50% and 36% in *Ppck-2* and *Ppck-4* lines, respectively. The quality of seeds was lower in *Ppck* lines: lower amino acid content, including Lys, and higher phytate content. These data confirm the relevance of the phosphorylation of PEPC in sorghum development, stress responses, yield, and quality of seeds.

## 1. Introduction

Phosphoenolpyruvate carboxylase (PEPC; EC 4.1.1.31) is a key enzyme in the metabolism of carbon and nitrogen, with central roles in photosynthesis, respiration, amino acid synthesis, and development and germination of seeds. This enzyme catalyzes the addition of bicarbonate to PEP to form oxaloacetate, which is reduced to malate by the enzyme malate dehydrogenase (MDH). PEPC is mostly recognized by its role in C_4_ and CAM photosynthesis [1,2], though it has also key functions in C_3_ plants and C_3_ tissues, such as seeds, fruits, roots, stomata, legume nodules, and others [3]. *Sorghum bicolor* PEPC gene family (*PPC* genes) include five plant-type PEPC genes (Sb*PPC1-5*: Sb10g021330, Sb02g021090, Sb04g00872, Sb07g014960, and Sb03g035090) and one bacterial-type PEPC gene (Sb*PPC6*, Sb03g0084810) [4].

PEPC is subjected to different post-translational modifications (PTMs), such as phosphorylation, monoubiquitination, NO-related PTMs (S-nitrosylation, Tyr-nitration), and carbonylation [5,6,7,8,9]. In addition, PEPC from *Arabidopsis thaliana* has been reported to be persulfidated in a proteomic assay [10]. These PTMs regulate PEPC activity and the turnover of the protein [9], although the biological function of some of them is not yet fully understood.

Protein phosphorylation is the best-known PTM of PEPC. This enzyme is phosphorylated at a conserved N-terminal serine residue by PEPC kinase (PEPCk), which belongs to the Ca^2+^/Calmodulin-dependent CDPK-SnRK protein kinase superfamily, although its activity is not Ca^2+^-dependent [11]. The phosphorylation of the photosynthetic isozymes occurs during the light period in C_4_ plants [12] and throughout the dark period in CAM plants [2], which corresponds with the pattern of *PPCK* gene expression. The phosphorylation of PEPC positively affects its functional and regulatory properties, decreasing its sensitivity to feedback inhibition by L-malate and increasing its affinity for the allosteric activator glucose-6-phosphate and Vmax [5]. The relevance of this PTM has been questioned because *Flaveria bidentis* RNAi lines, in which PEPC was not phosphorylated in the light, showed no differences in the CO_2_ assimilation rates with respect to the wild type [13]. On the contrary, a PEPCk-deficient *Kalanchoë fedtschenkoi* RNAi line showed perturbations of CAM photosynthesis, carbohydrate metabolism, and circadian rhythms [14]. *Sorghum bicolor* Sb*PPCK* gene family includes three genes (Sb*PPCK1-3*; Sb04g036570, Sb04g026490, and Sb06g022690) [15]. Zm*PPCK1* expression is triggered by light in mesophyll cells from maize, and its transcript is more abundant in mesophyll than bundle sheath cells [16]. For these reasons, it is considered the photosynthetic isoform. In sorghum, Sb*PPCK1* expression is higher in leaves than in roots, and is increased in the light; meanwhile, Sb*PPCK2* and Sb*PPCK3* are similarly expressed in leaves and roots [6].

Most stressful conditions affect the photosynthetic activity of plants, thus causing C shortage. In these situations, PEPC activity can contribute to improving C balance. There are many reports that show the relevant role of PEPC in water stress; salinity; N, P, and Fe deficiency; ammonium, Al, and Cd toxicity; and other abiotic stresses [17]. In many of these situations, increased PEPC activity was accompanied by a rise in PEPCk activity, as is summarized below.

Salinity is the major environmental factor restraining plant growth and productivity, and C_4_ plants are recognized as salt-tolerant plants. In sorghum, both PEPC and PEPCk proteins were conserved during prolonged salinity [18]. The main effect of salt stress in sorghum was a marked increase in leaf PEPCk activity [19], mainly as a consequence of decreased PEPCk protein degradation due to negative control of the ubiquitin–proteasome pathway by the phytohormone abscisic acid (ABA) [20]. The increase in the phosphorylation state of PEPC caused augmented malate synthesis, and it could contribute to alleviating carbon deficiency under salinity. In sorghum roots, salinity increased PEPC and PEPCk activity, and Sb*PPC3* and Sb*PPCK2* gene expression [21]. Likewise, salinity increased At*PPC3* expression, PEPC activity and its phosphorylation state, and L-malate content in Arabidopsis roots [22].

Despite being a common intermediate in plant metabolism, ammonium can produce toxicity symptoms in plants, causing leaf chlorosis, net photosynthesis decrease, and a reduction in growth [23]. PEPC activity, by supplying carbon skeletons for ammonium assimilation, plays a pivotal role in tolerance to ammonium stress. Ammonium increased sorghum PEPCk activity and the phosphorylation state of PEPC in leaves and roots [6]. In leaves, increased PEPCk activity was associated with PPCK1, the photosynthetic isozyme. In roots, increased PEPCk activity was dependent on the PPCK2 isozyme.

Orthophosphate (Pi) is an essential macronutrient required for many fundamental processes in plants, including photosynthesis and respiration, as well as nucleic acid, protein, and membrane phospholipid synthesis. Nevertheless, it is often a limiting nutrient in soils, and P deficiency is one of the greatest limitations in agricultural production [24]. To overcome low Pi availability, plants have evolved an array of responses. Among them, excreting organic acids, mainly malate and citrate, to acidify the rhizosphere and chelate metal ions has been related to increased PEPC [3,25] and PEPCk [26] activities. In sorghum, we have found that P deficiency increased the phosphorylation of PEPC in leaves, and both PEPC activity and PEPCk activity in roots. In Arabidopsis, phosphate starvation increased the expression of all At*PPC* and At*PPCK* genes in shoots, but only and specifically At*PPC3* and At*PPCK2* in roots [22].

Root synthesis and the excretion of organic acids also have relevant roles in Al and Cd tolerance. As in P deficiency, the role of PEPC contributing to organic acid synthesis is well documented. For example, a transgenic rice overexpressing C_4_-PEPC showed increased tolerance towards Al toxicity [27]. In sorghum, Al increased the synthesis of malic acid, more in tolerant than in Al-sensitive cultivars [28]. With respect to Sb*PPCK* gene expression, Al treatment increased Sb*PPCK2* and Sb*PPCK3* expressions in roots, and Sb*PPCK1* in leaves [29]. Similarly, Cd stress enhanced the activity of PEPC in several plant species (wheat, bean, maize). Organic acid anions are important chelators of toxic metals, including cadmium. In sorghum, Cd increased the expression of Sb*PPCK2* in roots and leaves [29]. Likewise, in Arabidopsis, Cd increased PEPC activity and its phosphorylation, and At*PPC* and At*PPCK* gene expression [30].

There is experimental evidence that sorghum PEPC is phosphorylated both in developing and germinating seeds [8,31]. Seed formation needs very active primary C/N metabolisms to build up structures, including embryo formation and endosperm cellularization, together with storage of stock resources for the forthcoming germination. Due to the key role of PEPC in C and N metabolisms, an impact of the lack of PEPC or PEPCk on the ability of seeds to germinate is expected. Arabidopsis SALK lines deficient in At*PPCK1 (ppck1^−^)* or At*PPCK2 (ppck2^−^)* showed reduced weight and delayed flowering [22]. Seeds from these plants had a marked deficiency in global seed nitrogen, and changes in lipid quantity and identity [32]. This shows that PEPC and its phosphorylation state impacts not only seed production but also its composition, modifying its value for human and animal food.

This study aimed to assess the impact of decreased PEPC kinase activity, and subsequent reduced phosphorylation of PEPC, in sorghum. This has been carried over with sorghum lines in which the expression of all three Sb*PPCK* genes was silenced by RNAi interference (*Ppck* lines). The impacts of silencing on growth, stress responses, and yield were investigated. The impact on vegetative growth was evaluated by measuring growth and biomass production, and responses to light, salinity, and phosphate deficiency. The impact on reproductive growth and yield was evaluated by the time of flowering and by the number and weight of seeds. In addition, several parameters related to the metabolomics and nutritional quality of both forage and seeds have been quantified: N and C content, metabolic profiles of amino acids and fatty acids, and, in seeds, tannins and phytates. Despite the uncompleted silencing of Sb*PPCK* genes, silenced *Ppck* lines showed alterations in metabolomics, growth, stress responses, flowering time, and amount and quality of seeds.

## 2. Results

### 2.1. Effects of SbPPCK1-3 Silencing on SbPPCK Gene Expression and PEPCk Activity

The effect of silencing on Sb*PPCK* gene expression was measured in leaves of T2 homozygous lines *Ppck-2* and *Ppck-4* (Figure 1). PPCK1 is the C_4_ photosynthetic isozyme, expressed in leaves but not in roots. Sb*PPCK1* gene expression was reduced by about 65% and 83% in *Ppck-2* and *Ppck-4* illuminated leaves, respectively. In addition, the expression of this gene significantly decreased by 25% in leaves of *Ppck-2* and *Ppck-4* kept in the dark. The impact of Sb*PPCK1-3* silencing was probably higher in illuminated leaves, due to the remarkable enhancement of Sb*PPCK1* expression by light (15-fold with respect to dark). Sb*PPCK2* is expressed in both leaves and roots. Because Sb*PPCK1* does not express in roots, and the expression of Sb*PPCK3* is very low in this tissue, PPCK2 is the main isozyme accounting for root PEPCk activity. Root Sb*PPCK2* gene expression was decreased by about 50% in *Ppck-2* and *Ppck-4.* Finally, the Sb*PPCK3* level of expression was low and highly variable in both leaves and roots.

The in vitro PEPCk activity in leaf extracts was quantified by the Pro-Q/SYPRO method (Table 1). This method allows the fluorescent detection of phosphoproteins directly in SDS-PAGE gels. Fluorescence signal intensity correlates with the number of phosphorylated residues, and the results are normalized to total protein with SYPRO Ruby gel stain. Protein extracts containing PEPCk were incubated with purified dephosphorylated sorghum C_4_ PEPC and ATP. The amount of phosphorylated exogenous PEPC at the end of the incubation period, referred to as the amount of protein in the same band, is considered the in vitro PEPCk activity. Similar activities were measured in Wt and *Ppck* lines. This indicates that the partial silencing of Sb*PPCK1-3* may not have a great impact on PEPCk activity. We cannot rule out that the incomplete silencing of Sb*PPCK* genes could allow the synthesis of similar quantities of PEPCk protein in Wt and *Ppck* lines. However, the phenotype found in *Ppck* lines indicates that the partial silencing of Sb*PPCK* genes in these lines might be enough to affect the PEPCk activity *in planta*.

### 2.2. Impact of SbPPCK1-3 Silencing on Phosphorylation of PEPC and Vegetative Development

Plant growth and biomass production and the phosphorylation state of PEPC were assessed in different conditions that have been associated with increased PEPCk activity, such as salinity and phosphate deficiency. In control conditions, *Ppck-2* length, shoot weight, and root weight were lower than in Wt (about 60, 30, and 40% of Wt, respectively) (Table 2). Salinity decreased these parameters to the same extent in the three lines: by 60% plant length and by 90% and 88% shoot and root weight. Due to the reduction in the growth of *Ppck-2* in the control condition, plants of this line were the smallest under salinity (Table 2).

The phosphorylation of PEPC in sorghum leaves was estimated with the malate test (IC_50_, malate concentration that causes 50% of inhibition of PEPC activity at pH 7.3). The IC_50_ of the inhibition of PEPC activity by malate is higher when PEPC is phosphorylated and lower for dephosphorylated PEPC. It reflects the in vivo PEPCk activity. Light increased the phosphorylation of PEPC in Wt, both in control and NaCl-treated plants (Figure 2). This effect was lost in *Ppck-2,* and greatly reduced in *Ppck-4*.

Phosphorous deficiency is known to increase root PEPC activity and PEPC phosphorylation [3,25,26]. The effect of P deficiency on plant growth and PEPC phosphorylation was evaluated in the absence of P (-P) and with treatment with insoluble P (calcium phosphate, PCa) due to the known role of PEPC in the synthesis and secretion of root exudates that increase P availability. In the absence of P, growth and biomass decreased, and the treatment with PCa partially recovered control values (Table 3). No remarkable differences were observed between lines, although lower values were measured in *Ppck-2*.

P deficiency increased PEPC activity in Wt and *Ppck-2* roots (Figure 3a), and this increase was higher in *Ppck-2*. This result shows that the treatment was effectively triggering a response in the plant, and the higher increase in PEPC activity in *Ppck-2* leaves may reflect compensation for the lack of phosphorylation of the protein. In leaves, P deficiency increased the phosphorylation of PEPC (IC_50_ malate) in Wt, but not in *Ppck-2* (Figure 3b).

Taken together, these results show that Sb*PPCK* gene silencing, although without great effect on growth, significantly decreased the phosphorylation of PEPC associated with stress responses.

### 2.3. Metabolic Changes in Ppck Lines

The effect of Sb*PPCK* gene silencing on the amino acid profile was investigated in leaves of sorghum lines after acid hydrolysis (Table 4). A general decrease in amino acid quantities was observed in *Ppck* lines, more notably in *Ppck-4* (Table 5). The amounts of Arg, Lys, and Val were significantly decreased by 8%, 7%, and 16% in *Ppck-2* and by 15%, 14%, and 18% in *Ppck-4*, respectively. Other amino acids were also significantly lower in *Ppck4* (Glu, Phe, His, Leu, Met, Ser). Total amino acid content was 4% and 14% lower in *Ppck-2* and *Ppck-4* lines, respectively (Table 5). In *Ppck-4*, the C/N ratio was significantly higher than Wt due to increased C and decreased N content (Table 5).

Leaf fatty acids were about 50% saturated, 15% monounsaturated, and 35% polyunsaturated in Wt plants. The amount of several fatty acids was increased in *Ppck* lines (palmitoleic, suberic, azelaic, α-linolenic, and arachidonic acid), and the amount of other acids was decreased (oleic and eicosenoic acid) (Table 6). Globally, a tendency towards decreased amounts of monounsaturated fatty acids in *Ppck* lines was observed (Figure 4).

### 2.4. Impact of SbPPCK1-3 Silencing on Reproductive Development and Seed Production

The time of flowering was delayed in *Ppck* lines (Figure 5). On day 49, 50% of Wt plants had flowers, but only 3% of *Ppck-2* and 0% of *Ppck-4*. At day 56, 100% of Wt plants had flowers, but only 66% of *Ppck-2* and 50% of *Ppck-4*. In addition, about 20% of *Ppck-4* plants did not produce flowers at all. In summary, 100% of Wt plants had flowers on day 56, 100% of *Ppck-2* plants on day 66, and 80% of *Ppck-4* plants on day 73. Panicle length and weight and seed weight and number were decreased in *Ppck* lines (Table 7). *Ppck-2* number of seeds was 50% of Wt. In *Ppck-4* plants, it was 60% of Wt in plants with flowers, but it is worth noting that, as mentioned above, 20% of plants of this line did not form flowers. These results clearly show that PEPC phosphorylation has a great impact on sorghum reproductive development and productivity.

### 2.5. Impact of SbPPCK1-3 Silencing on the Composition of Seeds

Seed composition was analyzed first to clarify the impact of Sb*PPCK* gene silencing on seed metabolites and second to evaluate changes related to the nutritional quality of seeds. The quantity of all the measured amino acids, except Met, was decreased in *Ppck* lines (Table 8). Accordingly, total amino acid content was lowered by 34% and 55% in these lines (Table 9). Meanwhile, starch was only decreased by 12% in *Ppck-4* (Table 9). On the contrary, two parameters that affect the nutritional quality of seeds (tannins and phytates) were increased in *Ppck* lines (Table 9). Although both total C and total N were lower in *Ppck* lines, the ratio C/N was higher in these lines due to a lesser N content (Table 10).

Fatty acid composition was different in seeds in comparison to leaves, with similar amounts (30–35%) of saturated, monounsaturated, and polyunsaturated fatty acids in seeds of Wt. Palmitic, stearic, γ- and α-linolenic, and arachidonic acids were increased, and oleic, arachidic, and eicosenoic acids were lowered in *Ppck* lines (Table 11). The same than in leaves, it was observed a tendency to a diminution of monounsaturated fatty acids (Figure 6).

These results show that Sb*PPCK* gene silencing negatively disturbs the reproductive development of sorghum, delaying flowering, decreasing seed production, and changing their metabolic profiles. In this respect, the highest alterations were produced in N compounds, with a noticeable decrease in total amino acid content in *Ppck* seeds.

## 3. Discussion

There has been controversy about the significance of PEPC phosphorylation in different physiological contexts [13,14]. In this work, we show that Sb*PPCK* gene silencing noticeably affects sorghum vegetative development, leaf and seed metabolic profiles, reproductive development, yield, and quality of seeds. We previously silenced *SbPPC3* using similar research tools and obtained several *Ppc3* lines with 80–90% *PPC3* gene silencing. In this work, the degree of silencing was smaller, about 25% in darkened leaves and 50% in roots. Nevertheless, a clear impact on PEPC phosphorylation was produced in conditions in which PEPCk activity increases, such as light, salinity, and P deficiency. In all these physiological contexts, the phosphorylation state of PEPC was lower in *Ppck* lines, and these lines showed altered vegetative growth. Although the impact of the silencing on biomass production was small, the reduced capacity of *Ppck* lines to cope with stress along development may contribute to the decreased yield of these lines, in addition to specific roles that PEPC phosphorylation may have in flowering and seed development. The results of this work highlight the relevance of PEPC phosphorylation for adequate growth and development of sorghum and its impact on sorghum productivity, and they are in accord with previous results of our group in Arabidopsis [22,32].

The key role of PEPC in C and N metabolisms is well documented [33,34]. The impact of Sb*PPCK* gene silencing was higher on N metabolism than on C metabolism. Both in leaves and in seeds, the ratio C/N increased in *Ppck* lines due to a reduction in total N, this effect being more remarkable in seeds. The amount of total amino acids was likewise higher in seeds than in leaves, and it was reduced to a greater extent by Sb*PPCK* gene silencing. Like maize, the grain of sorghum is deficient in essential amino acids, such as Lys [35]. Interestingly, Lys was 0.08% of leaf amino acids and 0.03% of seed amino acids, and these quantities were further reduced in *Ppck* lines. As Lys content affects nutritional quality, this result suggests that the nutritional quality of forage would be reduced in these lines. In sorghum seeds, the prolamins, referred to as kafirins, account for 70% of the total protein and are devoid of Lys [36]. It would be interesting to study whether Sb*PPCK* gene silencing increases the amount of kafirins, thus decreasing the total Lys level.

Two genes (At*PPCK1* and At*PPCK2*) encode PEPCks in *Arabidopsis thaliana* [37]. At*PPCK1* is the main PPCK gene expressed in Arabidopsis leaves [38] and transcripts of both At*PPCK1-2* accumulated at higher levels in roots [22]. SALK T-DNA knockout mutant *ppck1^−^* showed decreased yield and a marked N-deficiency, while *ppck2^−^* had lower levels of soluble proteins [32]. Both mutants had reduced pools of Krebs cycle intermediates, reflecting negative effects on anaplerotic metabolism that adversely affect amino acid synthesis and nitrogen metabolism. Leaves notably contribute to N-filling of the seeds [39,40], although seed filling and development depend partially on shoots and roots. In Arabidopsis, the *ppck1* genotype was mainly related to leaf metabolism, and *ppck2^−^* to root metabolism [32]. In this work, the effects of Sb*PPCK* gene silencing are expected to be dependent on both leaf and root metabolism, and related to reduced phosphorylation of photosynthetic and nonphotosynthetic PEPC isozymes. These combined effects importantly impact total seed amino acid content, greatly decreasing the nutritional value of these seeds. It is worth noting that plant metabolism controlling seed filling and quality is different between sorghum (starch-rich seeds) and Arabidopsis (oleaginous seeds). Nevertheless, the effect of the lack of PEPC phosphorylation has a comparable negative impact on N compounds of both types of seeds.

The metabolic profile of fatty acid was also modified in sorghum *Ppck* lines. Both in leaves and seeds, a tendency of decreased monounsaturated acid content was observed. This effect was more important in seeds, in which monounsaturated fatty acids were 30% of total fatty acids, while in leaves, they accounted for only 12% of total fatty acids. Arabidopsis *ppck^−^* mutants had some minor changes in lipid metabolism [32]. Seeds of the *ppck2^−^* Arabidopsis line had a higher polyunsaturated/no polyunsaturated ratio in comparison to Wt. In sorghum, this ratio was 0.57, 0.55, and 0.75 in Wt, *Ppck-2*, and *Ppck-4* seeds, respectively. Thus, the *Ppck-4* sorghum line had the same alteration as the Arabidopsis p*pck2^−^* mutant in seeds. It is well known that the fatty acid composition of seeds impacts their effects on human health [41]. This indicates that silencing of *PPCK* genes can change its nutritional quality by changing the fatty acid profile. Suberic (C8:0) and azelaic (C9:0) acids were measured because they have been found in some varieties of sorghum, products of linoleic acid endogenously formed by the peroxisomal ω-oxidation pathway. These clinically important saturated fatty acids can be additional sources of edible oil with anti-inflammatory and anti-oxidation properties [42]. In this work, we found that the amount of both acids was increased by 2-fold in *Ppck* lines with respect to Wt. Nevertheless, the amount of azelaic and suberic acids was low in Wt (0.16% and 0.18%, respectively), and these changes are probably of low relevance. In seeds, suberic acid was not detected, and the amount of azelaic acid was very low (0.05%) and similar in Wt and *Ppck*-lines.

The most remarkable effects of Sb*PPCK* gene silencing were the delayed flowering time and the reduction in seed production in sorghum. In Arabidopsis, the knockout of any At*PPC* gene delayed flowering [22], and seed yield was found to be highly reduced [32], especially in the p*pck1^−^* plant. In sorghum, delayed flowering and decreased yield were also observed in plants where the Sb*PPC3* gene had been silenced (*Ppc3* line) [21]. These observations indicate that all PEPC and PEPCk isozymes are needed for correct reproductive development and that phosphorylation of PEPC is essential in this respect. In addition, it affects the quality of seeds in several aspects. Sorghum grain is a major staple food for humans and a feed source for livestock in many regions of the world. Sorghum, like maize, is a good source of nutrients and is drought-resistant and gluten-free. Sorghum has the highest concentration of phenolic chemicals in any cereal grain. In vitro research suggests that sorghum phenolic compounds are antioxidants and positively impact human health (nutraceuticals) [43]. In comparison to other cereals, sorghum grains include tannins with a high molecular weight and a high degree of polymerization, with levels ranging from 10 to 68 mg^−1^ g dry weight, depending on the cultivar [44]. Nevertheless, the sorghum seeds used in this work had a very low amount of tannins, which was increased in *Ppck-2* plants. Although this increase is probably of low consequence, this result shows that silencing Sb*PPCK* genes may constitute another way to modify the nutritional quality of seeds. Similarly, phytate content was significantly increased in both *Ppck* lines. Phytate is the major storage compound for phosphorus in seeds and contributes to 1–5% of the total seed weight. It is widely regarded as an anti-nutrient in human diets and animal feed because it chelates metal ions, reducing their bioavailability and thus the nutritional value of seeds [45]. In addition, phytate forms complexes with basic amino acids, seed storage proteins, and enzymes in the animal digestive tract, which may reduce amino acid availability, protein digestibility, and the activity of digestive enzymes. Thus, Sb*PPCK* gene silencing reduces the quality of sorghum seeds in this respect.

On the whole, sorghum *Ppck* lines showed alterations in vegetative and reproductive development, and decreased seed quantity and quality. The silencing affects seed amino acid and fatty acid profiles, and the amounts of nutraceuticals and anti-nutrients. There are some differences between the two lines. A higher degree of silencing of Sb*PPCK1* and Sb*PPCK2* in illuminated leaves was observed in the *Ppck4* line. Meanwhile, Sb*PPCK1* and Sb*PPCK2* were equally silenced in darkened leaves and roots in both *Ppck* lines. It could be conjectured that those phenotypic characteristics more pronounced in the *Ppck4* line are related to the phosphorylation of photosynthetic PEPC (SbPPC1) and other non-photosynthetic leaf isozymes with anaplerotic functions. These effects are mainly a reduction in leaf and seed amino acid content and an alteration of the C/N ratio in both organs, together with delayed flowering. On the other hand, in the *Ppck-2* line, the lack of phosphorylation of root PEPC would be responsible for decreased biomass production (shoots, roots, and seeds) and delayed flowering. SbPPC3 is the main PEPC isozyme responsible for root PEPC activity in sorghum [21]. Accordingly, Sb*PPC3* silencing caused decreased root and shoot biomass production, a delay in flowering time, and reduced seed production in the *Ppc3* plant. Little is known about Sb*PPCK3*’s pattern of expression and physiological functions. Maize has four *PPCK* genes (Zm*PPCK1-4*). A higher degree of homology was found between Sb*PPCK3* and Zm*PPCK4* (GRMZM2G049541) [4]. In silico analysis of the gene expression pattern, using the ePlant Maize viewer public transcriptome repository suit [46], showed that Zm*PPCK4* is expressed in flowers (ears, tassels, and anthers) and seed endosperm [47]. It would be interesting to analyze whether the pattern Sb*PPCK3* and Zm*PPCK4* expression are similar, the role of Sb*PPCK3* in flower and seed development, and the impact that its silencing has on the reproductive development of sorghum.

In summary, this study aimed to assess the impact of reduced PEPCk activity and decreased phosphorylation of PEPC in sorghum. This was carried out with sorghum lines in which the expression of all three Sb*PPCK* genes was silenced by RNAi interference (*Ppck* lines). The degree of silencing was not 100%, but we have demonstrated that it decreased the degree of phosphorylation of PEPC in several physiological contexts (light, salinity, P deficiency). In these situations, the lower amount of Sb*PPCK* mRNA was associated with a decreased degree of PEPC phosphorylation. Decreased phosphorylation of PEPC, in turn, negatively affected sorghum growth and yield. This is important because there are scarce experimental results using PEC kinase mutants to demonstrate how this specific post-translational modification (phosphorylation) regulates in vivo PEPC activity and its impact in different physiological contexts. The silencing severely affects mainly two matters. First, it disturbed N metabolism, which caused a remarkable descent in amino acids in leaves and seeds. Second, it delayed flowering time and decreased yield. To a lesser extent, other features (fatty acids, vegetative growth, stress responses, quality of forage and seeds) were also altered in *Ppck* lines. Globally, the results of this work show the relevance of the phosphorylation of PEPC for adequate growth and yield of sorghum.

## 4. Materials and Methods

### 4.1. Plant Material and Growth Conditions

Sorghum (*Sorghum bicolor* L.) Wt and *Ppck* plants used in this study correspond to the public genotype P898012. Seeds were surface sterilized with 50% (*v/v*) bleach and 0.1% Triton X-100 for 30 min and rinsed 8–10 times with sterile water. Seeds were placed in moist sterile filter papers for 3 days in darkness at 25 °C. For hydroponic experiments, 3 seedlings were transferred to 1 L polyethylene pots filled with nitrate-type nutrient solution [48] and grown for 3 weeks in a growth chamber in 12 h light/dark cycles (25 °C, 60% relative humidity, and 20 °C, 70% relative humidity for each photoperiod, respectively), at 350 μmol m^−2^ s^−1^ PAR light intensity. For light/dark experiments, fully expanded youngest leaves were excised and illuminated for 2 h at 700 μmol m^−2^ s^−1^ PAR light intensity or kept in darkness. For NaCl treatments, after transferring the seedlings to the hydroponic cultures, plants were grown with nitrate-type nutrient solution with increasing amounts of NaCl: 0 mM (7 days), 86 mM (3 days), 172 mM (7 days), and 257 mM (5 days). For phosphorous deficiency experiments, plants were grown hydroponically with Hewitt medium (control), without phosphate (-P), or with 0.67 mM Ca_3_(PO_4_)_2_ (PCa) (as a source of insoluble phosphate) for 3 weeks. All samples were harvested within the first 2 h of the light period. For results in Table 4, Table 5 and Table 6 and Figure 4, plants were grown in soil pots for 4 weeks in a growth chamber with the same light, temperature, and humidity conditions as stated above, and leaves were harvested to carry out the analysis. For results in Figure 5 and Figure 6 and Table 7, Table 8, Table 9, Table 10 and Table 11, seedlings were transferred to soil pots and grown in a greenhouse (CITIUS, University of Sevilla, Sevilla, Spain) until obtaining the seeds 6 months later. To block cross-fertilization between the different lines, panicles were covered with paper bags.

### 4.2. Generation of Sorghum Ppck Lines

As described in [21], the binary vector pFGC161 was used to construct the hpRNA cassette. The T-DNA of the vector contained the selection gene bar, for resistance to the herbicide ammonium glufosinate, and the silencing cassette with two restriction sites separated by a rice waxy intron driven by the CaMV35S promoter. As we intended to silence all 3 *PPCK* genes (Sb*PPCK1-3*), and due to the high sequence homology between the 3 genes, we inserted the full coding sequence (CDS) of Sb*PPCK1* (855 bp) in the sense and anti-sense orientations in the hpRNA cassette. This fragment was amplified by PCR using the high-fidelity Q5 DNA polymerase (New England Biolabs) and cDNA from sorghum leaves, and purified by gel purification. The sequences of the primers used were (forward) 5′- ATACTAGTGGCGCGCCATGAGCGGCGCCGCG-GAG-3′ and (reverse) 5′- CCGAGCTCGCCTAGGTCAGGCCACCGCCACACT-3′. The fragment was inserted in sense and anti-sense orientation by AscI and AvrII, and by SacI and SpeI restriction enzymes, respectively. The ligation product was used to transform the DH5α strain of *Escherichia coli* by heat shock, and single colonies were selected on LB containing kanamycin as a selective marker. Correct cloning was confirmed by restriction analysis and sequencing. A schematic structure of the T-DNA of the vector pFGC161-Sb*PPCK* is presented in Figure 7.

The insertion of the vector pFGC161-Sb*PPCK* into *Agrobacterium tumefaciens* strain AGL1 and the transformation of immature embryos from the sorghum P898012 genotype were carried out at the Plant Transformation Core Facility, University of Missouri (Columbia, MO, USA) (https://research.missouri.edu/plant-transformation (accessed on 22 June 2023)), following the method described by Do et al. [49]. A total of 12 T0 plants from independent insertion events were obtained, named *Ppck-1-12* lines, and grown until obtaining seeds (T1).

### 4.3. Selection of Transformed Ppck Plants

In total, 20 T1 seeds from each independent line were sown in soil pots, and the insertion of the T-DNA for Sb*PPCK1-3* silencing was confirmed by herbicide ammonium glufosinate application to leaves and by PCR for the presence of waxy and bar genes in the genome [21]. PCR reactions were carried out using 200 ng of gDNA, iTaq DNA polymerase (Intron Biotechnology), and the specific primers (barF: 5′-AAACCCACGTCATGCCAGTT-3′, barR: 5′- CATCGAGACAAGCACGGTCA-3′; waxyF: 5′-GTAGCCGAGTTGGTCAAAGGA-3′, waxyR: 5′-TTCTTGGGTGGCTAGGGGATA-3′). gDNA was extracted from 100 mg of shoot tissue from seedlings using the i-genomic plant DNA extraction kit (Intron Biotechnology, Burlington, MA, USA), following the manufacturer’s instructions. T1 plants showing no herbicide symptoms and the presence of waxy and bar genes were selected for Sb*PPCK1-3* expression analysis by qPCR.

Sb*PPCK1-3* silencing degree in transgenic Ppck T1 lines was evaluated by qPCR analysis as described in Arias-Baldrich et al. [6], using actin as standard (Figure 8). Among the 12 *Ppck* T1 lines, *Ppck-2*, *Ppck-3, Ppck-4*, and *Ppck-9* showed the highest degree of silencing for Sb*PPCK1-3* genes. Several parameters related to plant growth and productivity were decreased in *Ppck* lines (Table 12). All of them (plant length, panicle length and weight, seed weight and number) were markedly lower in *Ppck-9* with respect to Wt. All the parameters were also decreased in *Ppck-2* and, except plant length, in *Ppck-4*.

Taking into account the degree of Sb*PPCK* silencing and phenotypic variation, *Ppck-2*, *Ppck-4*, and *Ppck-9* lines were selected. These lines were self-fertilized to obtain homozygous T2 lines. Panicles were covered with paper bags to block cross-fertilization between *Ppck* lines. Seeds from Wt and *Ppck* plants used in this study were sown, grown, and harvested in the same conditions and time, corresponding to the same batch. All plants used in this work were confirmed by herbicide application and PCR, as described in the section above on the selection of transformed *Ppck* plants.

### 4.4. Site of Insertion of the T-DNA into Ppck Lines Genome

To determine the number of insertions and the place where the T-DNA had been inserted in the sorghum genome, we sequenced the whole genome of *Ppck-2*, *Ppck-4*, and *Ppck-9* plants from homozygous T2 lines (Novogene Europe). Genomic DNA (gDNA) used for sequencing was extracted from 50 to 100 mg leaf tissue using the i-genomic Plant DNA Extraction Mini Kit (iNtRON Biotechnology), following the manufacturer’s instructions. Prior to library construction, the quality of gDNA samples was tested by agarose gel electrophoresis and quantified by Qubit 2.0. To prepare the library, the gDNA was randomly fragmented by sonication, and then DNA fragments were end-polished, A-tailed, and ligated with the full-length adapters of Illumina sequencing, followed by further PCR amplification with P5 and indexed P7 oligos. The PCR products as the final construction of the libraries were purified with the AMPure XP system. Then, libraries were checked for size distribution by the Agilent 2100 Bioanalyzer (Agilent Technologies, CA, USA), and quantified by real-time PCR (to meet the criteria of 3 nM). The qualified libraries were fed into Illumina sequencers (Novogene Europe) for whole-genome sequencing (WGS). Genome analysis showed one T-DNA insertion for the *Ppck-2* line in a non-coding region (position 75.666.309) in chromosome 1 (NC_012870.2). This insertion was located at 2683 bp upstream of the 5′-end of a solute transporter (LOC8059798; XP_002468411) and 6882 bp downstream of the 3′-end of an uncharacterized protein (LOC110431802; XP_021307090). In the *Ppck-4* line, we found two different T-DNA insertions, one in chromosome 1 (NC_012870.2) and one in chromosome 2 (NC_012871.2). In both cases, the T-DNAs were inserted in non-coding regions (position 60.151.914 in chromosome 1, position 67.984.045 in chromosome 2). The insertion in chromosome 1 was located at 2231 bp upstream of the 5′-end of a peroxidase 5 gene (LOC8062867; XP_002467585) and 5120 bp downstream of the 3′-end of another peroxidase 5 gene (LOC8058813; XP_002467586). The insertion in chromosome 2 was located at 1920 bp upstream of the 5′-end of an asparagine-ARNt ligase gene (LOC8080686; XP_002460704) and 228 bp downstream of the 3′-end of a gene similar to WAT1 At5g4740 (LOC8077443; XP_021310390). Finally, *Ppck-9* genome analysis showed one T-DNA insertion in chromosome 5 (NC_012874.2); however, the insertion was detected interrupting an intron from a caffeoyl shikimate esterase gene (position 35.916). This protein participates in lignin biosynthesis [50]. Although the T-DNA was inserted in an intron and not in an exon, the marked phenotype found in *Ppck-9* plants could be, at least partially, due to the unintended interruption of this caffeoyl shikimate esterase gene. Therefore, for the rest of the analysis in this work, we discarded the *Ppck-9* line and focused on *Ppck-2* and *Ppck-4* lines.

### 4.5. Determination of Enzyme Activity and Protein Quantification

Protein extracts were obtained by grinding 0.2 g fresh weight of leaf or root tissue in 1 mL of extraction buffer containing 0.1 M Tris–HCl pH 7.5, 20 % (*v/v*) glycerol, 1 mM EDTA, 10 mM MgCl_2_, a protease inhibitor cocktail (Sigma), 10 mM potassium fluoride, and 14 mM β-mercaptoethanol. The homogenate was centrifuged at 15.000× *g* for 2 min and PEPC activity was quantified in the supernatant. Protein concentrations were determined using the method of Bradford [51] with BSA as a standard.

PEPC activity was measured spectrophotometrically at optimal pH 8.0 using the NAD-MDH-coupled assay at 2.5 mM PEP [52]. A single enzyme unit (U) is defined as the amount of PEPC that catalyzes the carboxylation of 1 µmol of phosphoenolpyruvate per minute at pH 8 and 30 °C.

PEPCk activity was measured by staining phosphorylated proteins with Pro-Q Diamond. The in vitro PEPCk activity of sorghum leaves or roots was measured in aliquots of protein extracts (25 µg) that were incubated in a reaction medium containing 100 mM Tris-HCl, pH 7.5, 20% (*v/v*) glycerol, 5 mM MgCl_2_, 0.25 mM P1P5-di(adenosine-5′)-pentaphosphate (adenylate kinase inhibitor), 1 mM EGTA, and 0.1 units of nonphosphorylated purified sorghum C_4_ PEPC. The phosphorylation reaction was initiated by the addition of 1 mM ATP. The reaction was stopped by boiling the samples for 5 min at 90 °C in the presence of a loading buffer. The denatured proteins were separated by SDS-PAGE. Fluorescent staining of SDS-PAGE gels was performed using Pro-Q Diamond phosphoprotein stain and SYPRO Ruby protein stain (Invitrogen-Molecular Probes, Carlsbad, CA, USA), following the manufacturer’s instructions. After electrophoresis, the gels were stained with Pro-Q Diamond phosphoprotein stain and analyzed with a phosphor imager (Fuji FLA-5100; Fuji, Tokyo), and then stained with SYPRO Ruby and re-imaged. The positive control for the Pro-Q Diamond stain was phosphorylated ovalbumin, which is included in molecular weight markers and is the single protein stained by Pro-Q in this lane.

### 4.6. RNA Extraction and cDNA Synthesis

Total RNA was extracted from 100 mg of frozen powdered leaves or roots using the Plant RNA Isolation Mini Kit (Agilent Technologies). Extracted nucleic acids were DNase-treated to wipe out genomic DNA. RNA concentration was determined using Nanodrop 2000 (Thermo). Reverse transcription reactions were performed using the Transcriptor First Strand cDNA Synthesis Kit (Roche) and HISCRIPT II 1st Strand cDNA Synthesis Kit (Vazyme), following the manufacturers’ instructions, with 1 µg of purified total RNA. cDNA synthesized was used for qPCR experiments.

### 4.7. Quantitative Real-Time PCR (qPCR)

Quantitative PCR reactions (qPCR) were performed in a final volume of 20 μL consisting of 1 μL of the cDNA, 15 μM of the specific primers, and 10 μL of SensiFAST SYBR No-ROX kit (Bioline). PCR was conducted on the Light Cycler 480 II Real-Time PCR System (Roche), and the threshold cycles (Ct) were determined using the Light Cycler 480 software for all treatments. To normalize the obtained values, actin was used as an internal control in each sample. Relative gene transcript abundance data were obtained through the Livak method [53] (2^−ΔΔCt^), assigning the relative value of 1 for transcript abundance in Wt leaves or roots in control conditions.

### 4.8. Quantification of Amino Acids

Lyophilized samples (0.1 g) were ground in liquid nitrogen with 1 mL of 6N HCl for hydrolyzing proteins. Then, they were dried at 100 °C and kept overnight in a rotary evaporator. Subsequently, they were resuspended in 1 mL of 70% methanol, incubated at 4 °C overnight, and the supernatant was analyzed by gas chromatography–mass spectrometry (GC-MS) using a Thermo Scientific instrument at the Mass Spectrometry Services (CITIUS, University of Sevilla, Seville, Spain).

### 4.9. Determination of C/N Ratio

Analyses of the total C and N were performed by a LECO Elemental CNS-Trumac Autoanalyzer, version 1.3x (2014), at the Mass Spectrometry Services (CITIUS, University of Sevilla).

### 4.10. Quantification of Starch

The quantification of starch was carried out by digestion with amylolytic enzymes (amyloglucosidase and amylase) and subsequent quantification of reducing sugars [54]. Previously, free sugars were removed with successive 80% ethanol washings. After digestion, an aliquot was assayed with 3,5-dinitrosalicylic acid (DNS) and Na/K tartrate measuring the absorbance at 570 nm in a VERSA max microplate spectrophotometer (Molecular Devices, San Jose, CA, USA), using glucose for the standard curve.

### 4.11. Quantification of Fatty Acids

Lyophilized samples (0.1 g) were ground in liquid nitrogen. Fatty acids (FAs) were quantified at the Agricultural Research Service (CITIUS, University of Sevilla). Fat extraction and direct methylation of FAs were performed in a single-step procedure based on the method by Sukhija and Palmquist [55]. Separation and quantification of FA methyl esters (FAMEs) were carried out using a gas chromatograph (Agilent 6890N Network GS System, Agilent, Santa Clara, CA, USA) equipped with a flame ionization detector (FID) and automatic sample injector HP 7683, and fitted with an HP-88 J&W fused silica capillary column (100 m, 0.25 mm intern diameter, 0.2 µm film thickness; Agilent Technologies Spain, S.L., Madrid, Spain). Nonanoic acid methyl ester (C9:0 ME) was used as an internal standard (Sigma Chemical Co., Ltd., Poole, UK). Individual FAs were identified by comparing their retention times with those of the authenticated standard FA mix Supelco 37 (Sigma Chemical Co., Ltd., Poole, UK).

### 4.12. Quantification of Tannins

The ferric reagent was used for the quantification of condensed tannins according to the method of Porter et al. [56]. Samples were tested with 2% ferric ammonium sulfate and butanol:HCl (95:5 *v/v*), and absorbance was measured at 550 nm in a VERSA max microplate spectrophotometer (Molecular Devices).

### 4.13. Quantification of Phytates

Phytates were quantified with the Phytic Acid Assay Kit (K-PHYT_Megazyme). The samples were mixed with 0.66 M HCl and incubated at room temperature overnight in darkness to extract phytic acid and other inositol phosphates. Samples were centrifuged and supernatants were neutralized by adding 0.5 mL of 0.75 M NaOH. The extraction of phytic acid was performed in duplicate, and from each of the neutralized samples, two replicate analyses were carried out. Samples were dephosphorylated or not in the presence (for total Pi measurement) or absence (for free Pi measurement) of phytase and alkaline phosphatase, respectively. Pi was quantified as phosphorus from a calibration curve generated using standards of known phosphorous concentrations. For all the samples, free and total phosphorus contents were determined spectrophotometrically (VERSA max microplate spectrophotometer) at 655 nm, and their difference was calculated for the phytic acid content (g/g).

### 4.14. Statistical Analysis

All data were analyzed by ANOVA. Then, differences between group means were compared by the Duncan or Games–Howell multiple range test. A *p* value of <0.05 was considered to be statistically significant. All analyses were conducted using SPSS Statistics 21 (IBM, Armonk, NY, USA). Different letters indicate statistically significant differences.

## 5. Conclusions

This work shows that Sb*PPCK* gene silencing negatively impacts sorghum growth, stress responses, reproductive development, yield, and quality of forage and seeds. As a whole, it highlights the key role of the phosphorylation of PEPC in all these physiological contexts.

## Figures and Tables

**Figure 1 plants-12-02426-f001:**
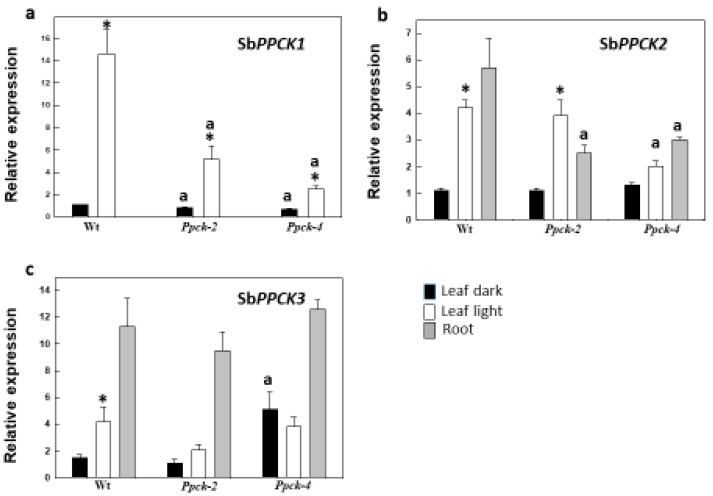
Expression of Sb*PPCK1* (**a**), Sb*PPCK2* (**b**), and Sb*PPCK3* (**c**) in leaves and roots of sorghum lines. Leaves were illuminated (2 h, 700 μmol m^−2^ s^−1^ PAR) or kept in the dark prior to RNA extraction and cDNA synthesis. Data are % of Wt in the dark. Data are means ± SE (*n* = 4). ^a^
*p* < 0.05 versus Wt. * *p* < 0.05 versus dark.

**Figure 2 plants-12-02426-f002:**
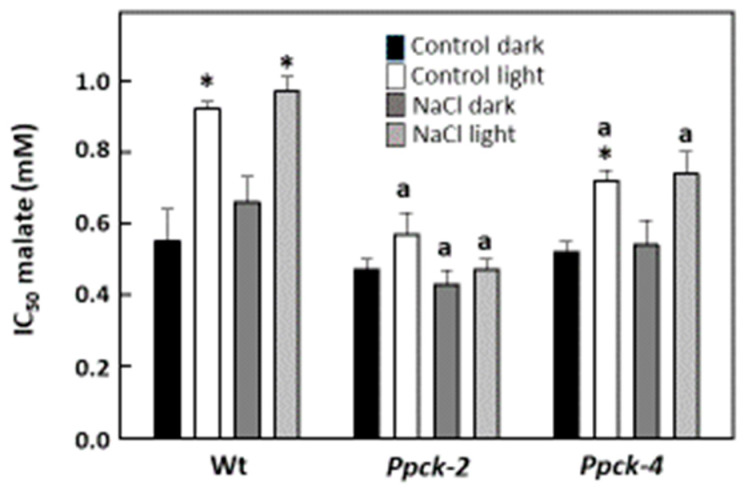
Effect of light and NaCl on the phosphorylation of leaf PEPC of sorghum lines. Plants were grown with increasing amounts of NaCl: 0 mM (7 days), 86 mM (3 days), 172 mM (7 days), and 257 mM (5 days), and measurements were made at the end of treatments. Leaves were illuminated (700 μmol m^−2^ s^−1^ PAR), or kept in the dark for 2 h before the preparation of protein extracts. Data are means ± SE (*n* = 3). ^a^
*p* < 0.05 versus Wt. * *p* < 0.05 versus dark.

**Figure 3 plants-12-02426-f003:**
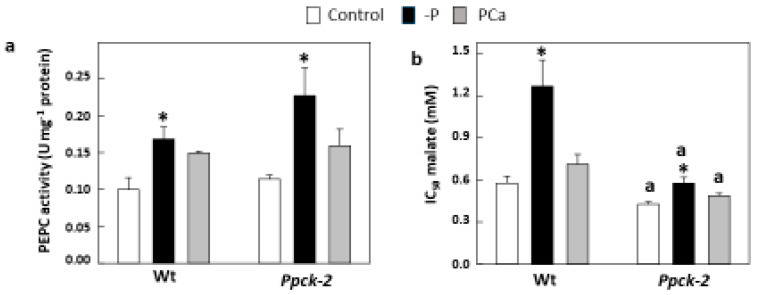
Root PEPC activity (**a**) and leaf PEPC phosphorylation (**b**) of sorghum plants with different phosphorous nutrition. Plants were grown for 3 weeks with Hewitt medium (control), without phosphate (-P), or with 0.67 mM Ca_3_(PO_4_)_2_ (PCa). Data are means ± SE (*n* = 3). ^a^
*p* < 0.05 versus Wt. * *p* < 0.05 versus Control.

**Figure 4 plants-12-02426-f004:**
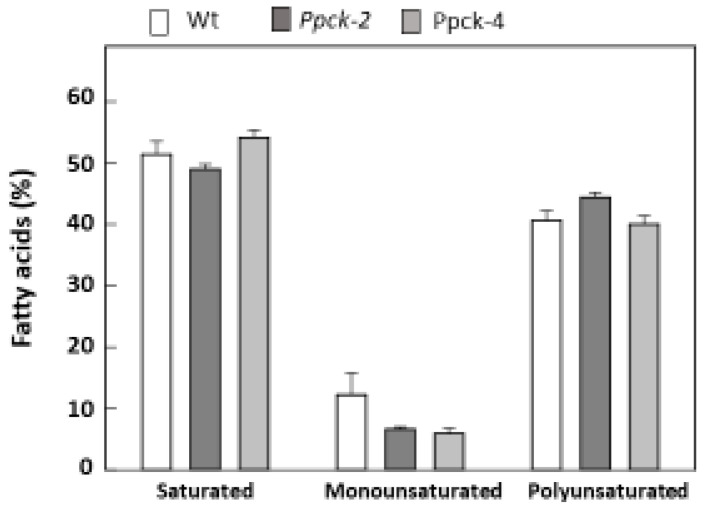
Total saturated, monounsaturated, and polyunsaturated fatty acids in sorghum leaves. Data are means ± SE (*n* = 3).

**Figure 5 plants-12-02426-f005:**
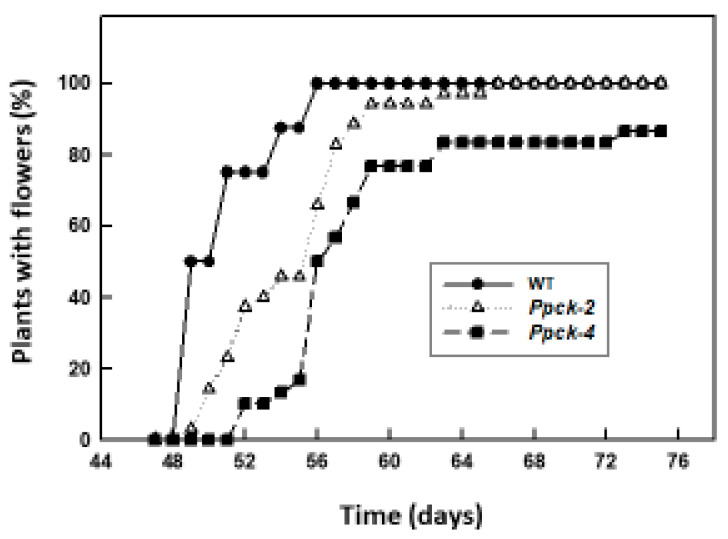
Flowering rate of sorghum lines. Data are % of plants with flowers at the indicated time.

**Figure 6 plants-12-02426-f006:**
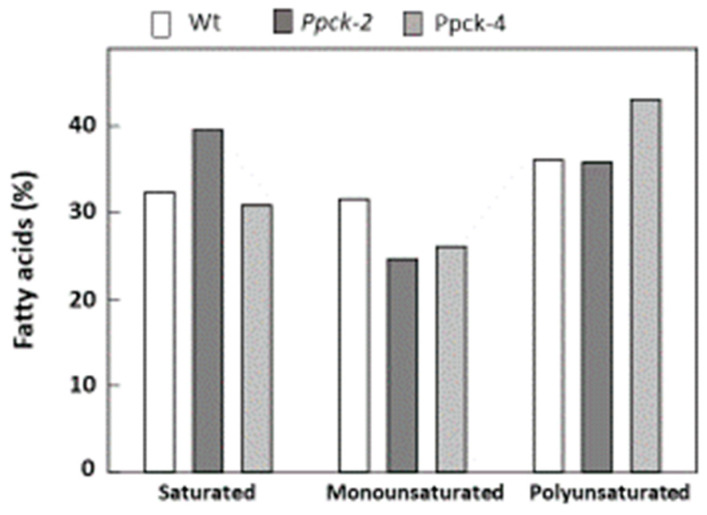
Total saturated, monounsaturated, and polyunsaturated fatty acids in sorghum seeds.

**Figure 7 plants-12-02426-f007:**
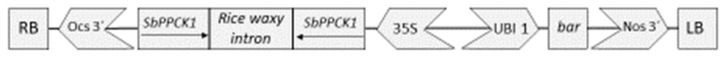
Structure of the T-DNA of the vector pFGC161-Sb*PPCK*.

**Figure 8 plants-12-02426-f008:**
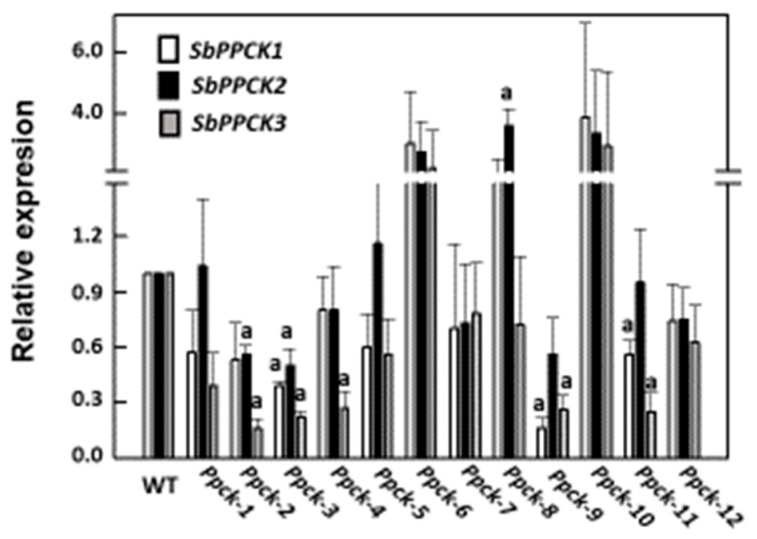
Relative level of expression of Sb*PPCK1-3* in leaves of *Ppck* lines. Data are means ± SE (*n* = 3). a, *p* < 0.05 versus Wt.

**Table 1 plants-12-02426-t001:** In vitro PEPCk activity of sorghum lines.

Line	Pro-Q/SYPRO (%)
Wt dark	100
Wt light	138 ± 22
*Ppck-2* dark	134 ± 29
*Ppck-2* light	141 ± 36
*Ppck-4* dark	119 ± 25
*Ppck-4* light	125 ± 24

PEPCk activity of illuminated leaves (2 h, 700 μmol m^−2^ s^−1^ PAR), or leaves kept in the dark, was measured in protein extracts (25 µg) with 0.1 U of purified C_4_ PEPC. Data are % of Wt in the dark. Data are means ± SE (*n* = 3).

**Table 2 plants-12-02426-t002:** Effect of salinity on growth and biomass production by *Ppck* lines.

Line	Treatment	Plant Length (cm)	Shoot Weight (g)	Root Weight (g)
Wt	Control	54.3 ± 0.5	5.4 ± 0.2	4.1 ± 0.3
	NaCl	20.9 ± 0.3 *	0.5 ± 0.1 *	0.5 ± 0.1
*Ppck-2*	Control	34.2 ± 3.6 ^a^	1.6 ± 0.1 ^a^	1.7 ± 0.2
	NaCl	12.4 ± 0.1 *^a^	0.2 ± 0.1 *^a^	0.2 ± 0.1
*Ppck-4*	Control	50.8 ± 1.1 ^a^	5.4 ± 0.7	3.8 ± 0.7
	NaCl	19.0 ± 2.4 *	0.4 ± 0.1 *	0.3 ± 0.1

Plants were grown with increasing amounts of NaCl: 0 mM (7 days), 86 mM (3 days), 172 mM (7 days), and 257 mM (5 days), and measurements were made at the end of treatments. Each sample was the mean of the measurements of the 3 plants of the same hydroponic. Data are means ± SE of 3 hydroponics (9 plants). ^a^
*p* < 0.05 versus Wt. * *p* < 0.05 versus Control.

**Table 3 plants-12-02426-t003:** Effect of phosphate deficiency on growth and biomass production by *Ppck* lines.

Line	Treatment	Plant Length (cm)	Shoot Weight (g)	Root Weight (g)
Wt	Control	61.5 ± 0.7	22.9 ± 1.8	11.5 ± 1.1
	-P	29.7 ± 3.1 *	3.0 ± 0.2 *	4.0 ± 0.1 *
	PCa	51.6 ± 2.0 *	12.1 ± 3.1 *	9.4 ± 2.2
*Ppck-2*	Control	56.6 ± 1.2 ^a^	22.6 ± 3.9	11.2 ± 1.4
	-P	26.5 ± 0.2 *	2.8 ± 0.1 *	3.8 ± 0.1 *
	PCa	48.8 ± 1.5 *^a^	10.8 ± 2.8	8.1 ± 1.4

Plants were grown for 3 weeks with Hewitt medium (control), without phosphate (-P), or with 0.67 mM Ca_3_(PO_4_)_2_ (PCa). Each sample was the mean of the measurements of the 3 plants of the same hydroponic. Data are means ± SE of 4 hydroponics (12 plants). ^a^
*p* < 0.05 versus Wt. * *p* < 0.05 versus Control.

**Table 4 plants-12-02426-t004:** Amino acid content in leaves of sorghum lines.

Amino Acid (mg g^−1^ fw)	Wt	*Ppck-2*	*Ppck-4*
Asp	1.82 ± 0.02	1.78 ± 0.04	1.57 ± 0.10
Glu	2.11 ± 0.02	2.00 ± 0.04	1.70 ± 0.08 ^a^
Ala	1.36 ± 0.05	1.40 ± 0.02	1.22 ± 0.08
Arg	1.01 ± 0.02	0.93 ± 0.02 ^a^	0.86 ± 0.05 ^a^
Cys	0.07 ± 0.01	0.06 ± 0.01	0.07 ± 0.01
Phe	1.28 ± 0.02	1.23 ± 0.02	1.14 ± 0.05 ^a^
Gly	1.04 ± 0.07	1.01 ± 0.06	0.97 ± 0.07
His	0.60 ± 0.01	0.57 ± 0.01	0.52 ± 0.03 ^a^
Ile	0.78 ± 0.01	0.75 ± 0.02	0.67 ± 0.03 ^a^
Leu	1.79 ± 0.03	1.70 ± 0.05	1.47 ± 0.10 ^a^
Lys	1.46 ± 0.02	1.36 ± 0.03 ^a^	1.25 ± 0.08 ^a^
Met	0.32 ± 0.01	0.34 ± 0.01	0.22 ± 0.03 ^a^
Pro	1.30 ± 0.01	1.29 ± 0.06	1.17 ± 0.02 ^a^
Ser	0.79 ± 0.01	0.78 ± 0.01	0.73 ± 0.04
Tyr	0.52 ± 0.01	0.49 ± 0.02	0.45 ± 0.04
Hyp	0.09 ± 0.01	0.09 ± 0.01	0.08 ± 0.01
Thr	0.91 ± 0.01	0.89 ± 0.01	0.81 ± 0.05
Val	0.89 ± 0.04	0.75 ± 0.01 ^a^	0.73 ± 0.01 ^a^

Data are means ± SE of 3 hydroponics (9 plants). ^a^
*p* < 0.05 versus Wt.

**Table 5 plants-12-02426-t005:** Total amino acid, C, and N content in leaves of sorghum lines.

Line	Amino Acids (mg g^−1^ fw)	Total C (%)	Total N (%)	C/N Ratio
Wt	18.10 ± 0.18	39.73 ± 2.58	3.92 ± 0.19	10.12 ± 0.26
*Ppck-2*	17.39 ± 0.40	42.93 ± 0.40	4.17 ± 0.07	10.30 ± 0.11
*Ppck-4*	15.62 ± 0.81 ^a^	42.55 ± 0.14	3.76 ± 0.08	11.32 ± 0.26 ^a^

Total amino acid is the sum of the values in Table 4. Data are means ± SE. ^a^
*p* < 0.05 versus Wt.

**Table 6 plants-12-02426-t006:** Fatty acid content in leaves of sorghum lines.

Fatty Acid (%)	Wt	*Ppck-2*	*Ppck-4*
Palmitic acid	26.88 ± 0.68	25.55 ± 0.38	28.01 ± 0.63
Palmitoleic acid	1.26 ± 0.04	1.55 ± 0.06 ^a^	1.59 ± 0.06 ^a^
Suberic acid	0.18 ± 0.01	0.29 ± 0.01 ^a^	0.34 ± 0.03 ^a^
Azelaic acid	0.16 ± 0.01	0.30 ± 0.02 ^a^	0.29 ± 0.02 ^a^
Stearic acid	12.54 ± 1.48	12.59 ± 0.25	14.17 ± 0.71
Oleic acid	7.73 ± 2.98	2.96 ± 0.39	2.29 ± 0.63
Linoleic acid	6.67 ± 0.59	8.17 ± 0.48	5.86 ± 0.22
γ-Linolenic acid	0.37 ± 0.05	0.26 ± 0.04	0.34 ± 0.04
Arachidic acid	0.53 ± 0.04	0.53 ± 0.07	0.60 ± 0.08
α-Linolenic acid	28.64 ± 3.61	34.85 ± 0.58	32.44 ± 1.06
Eicosenoic acid	0.74 ± 0.26	0.33 ± 0.04	0.17 ± 0.02 ^a^
Arachidonic acid	0.11 ± 0.02	0.25 ± 0.04 ^a^	0.33 ± 0.04 ^a^

Data are means ± SE (*n* = 3). ^a^
*p* < 0.05 versus Wt.

**Table 7 plants-12-02426-t007:** Seed production of sorghum lines.

Line	Panicle Length (cm)	Panicle Weight (g)	Seed Weight (g)	Seed Number
Wt	9.3 ± 0.8	6.9 ± 0.8	5.7 ± 0.6	204 ± 20
*Ppck-2*	7.3 ± 0.1 ^a^	4.0 ± 0.5 ^a^	3.4 ± 0.4 ^a^	101 ± 20 ^a^
*Ppck-4*	8.2 ± 0.6	5.2 ± 1.1	4.3 ± 0.6	131 ± 22

Data are means ± SE (*n* = 4) per panicle. ^a^
*p* < 0.05 versus Wt.

**Table 8 plants-12-02426-t008:** Amino acid content in seeds of sorghum lines.

Amino Acid (mg g^−1^ fw)	Wt	*Ppck-2*	*Ppck-4*
Asp	4.86 ± 0.40	3.45 ± 0.18 ^a^	2.44 ± 0.39 ^a^
Glu	16.01 ± 1.55	9.91 ± 0.62 ^a^	6.64 ± 1.12 ^a^
Ala	6.62 ± 0.46	4.23 ± 0.26 ^a^	2.89 ± 0.57 ^a^
Arg	2.56 ± 0.30	1.94 ± 0.02	0.29 ± 0.04 ^a^
Cys	0.54 ± 0.03	0.41 ± 0.01 ^a^	0.07 ± 0.01 ^a^
Phe	3.50 ± 2.61	2.55 ± 0.16 ^a^	1.85 ± 0.27 ^a^
Gly	2.30 ± 0.21	1.64 ± 0.10 ^a^	1.20 ± 0.22 ^a^
His	2.09 ± 0.17	1.49 ± 0.08 ^a^	1.01 ± 0.15 ^a^
Ile	2.38 ± 0.21	1.59 ± 0.06 ^a^	1.15 ± 0.19 ^a^
Leu	10.40 ± 0.95	6.76 ± 0.40 ^a^	4.25 ± 0.69 ^a^
Lys	1.95 ± 0.18	1.47 ± 0.08 ^a^	1.08 ± 0.17 ^a^
Met	0.43 ± 0.09	0.25 ± 0.02	0.28 ± 0.02
Pro	7.85 ± 0.60	5.02 ± 0.25 ^a^	3.81 ± 0.72 ^a^
Ser	2.99 ± 0.22	2.01 ± 0.13 ^a^	1.31 ± 0.20 ^a^
Tyr	0.82 ± 0.01	0.71 ± 0.03	0.48 ± 0.08 ^a^
Hyp	0.12 ± 0.01	0.11 ± 0.01 ^a^	0.09 ± 0.01 ^a^
Thr	2.23 ± 0.18	1.57 ± 0.07 ^a^	1.10 ± 0.18 ^a^
Val	3.42 ± 0.35	2.12 ± 0.18 ^a^	1.43 ± 0.25 ^a^

Data are means ± SE (*n* = 3). Samples were composed of homogenates of 5 g of seeds. ^a^
*p* < 0.05 versus Wt.

**Table 9 plants-12-02426-t009:** Effect of Sb*PPCK* gene silencing on seed composition.

Line	Amino Acids (mg g^−1^ fw)	Starch (µmol mg^−1^)	Tannins (mg g^−1^)	Phytates (g%)
WT	71.05 ± 6.13	1.43 ± 0.02	1.39 ± 0.36	4.37 ± 0.08
*Ppck-2*	47.23 ± 2.66	1.41 ± 0.03	2.36 ± 0.25 ^a^	4.78 ± 0.07 ^a^
*Ppck-4*	32.52 ± 5.46 ^a^	1.23 ± 0.04 ^a^	1.50 ± 0.36	5.47 ± 0.08 ^a^

Total amino acid is the sum of the values in Table 8. Samples were composed of homogenates of 5 g of seeds. Data are means ± SE (*n* = 3). ^a^
*p* < 0.05 versus Wt.

**Table 10 plants-12-02426-t010:** Total C and N content in seeds of sorghum lines.

Line	Total C (%)	Total N (%)	C/N Ratio
Wt	41.90	1.51	27.75
*Ppck-2*	40.64	1.32	30.81
*Ppck-4*	40.07	1.13	35.59

Samples were composed of homogenates of 5 g of seeds.

**Table 11 plants-12-02426-t011:** Fatty acid content in seeds of sorghum lines.

Fatty Acid (%)	Wt	*Ppck-2*	*Ppck-4*
Palmitic acid	19.20	24.81	19.91
Palmitoleic acid	0.49	1.24	0.16
Suberic acid	0.05	0.05	0.05
Azelaic acid	0.05	0.05	0.04
Stearic acid	5.69	11.09	7.40
Oleic acid	26.62	22.21	25.45
Linoleic acid	34.00	33.64	39.40
γ-Linolenic acid	0.16	0.20	0.31
Arachidic acid	0.16	0.00	0.00
α-Linolenic acid	1.52	1.43	2.40
Eicosenoic acid	1.31	0.18	0.07
Arachidonic acid	0.10	0.06	0.36

Measurements were performed in a mix of 5 g of powdered seeds.

**Table 12 plants-12-02426-t012:** Phenotypic characterization of *Ppck* lines.

Line	Plant Length (cm)	Panicle Length (cm)	Panicle Weight (g)	Seed Weight (g)	Seed Number
Wt	142.1 ± 3.2	13.5 ± 0.6	12.0 ± 2.6	9.9 ± 2.2	365.4 ± 64.7
*Ppck-1*	138.9 ± 3.2	14.4 ± 1.0	10.8 ± 0.8	9.9 ± 1.2	300.0 ± 48.0
*Ppck-2*	128.7 ± 4.2 ^a^	12.1 ± 0.7	9.2 ± 1.7	8.3 ± 1.4	271.2 ± 52.7
*Ppck-3*	119.7 ± 4.5 ^a^	13.8 ± 0.8	17.0 ± 2.2 ^a^	12.8 ± 2.3	478.9 ± 67.0
*Ppck-4*	150.8 ± 3.8	11.7 ± 0.3 ^a^	9.0 ± 0.8	8.2 ± 0.7	268.3 ± 23.7
*Ppck-5*	147.0 ± 7.6	13.2 ± 1.3	11.2 ± 2.2	13.0 ± 5.2	453.3 ± 133.1
*Ppck-6*	144.7 ± 4.5	11.4 ± 0.5 ^a^	9.4 ± 0.7	7.5 ± 0.8	245.8 ± 23.7
*Ppck-7*	137.8 ± 8.6	15.1 ± 0.7	18.2 ± 3.0	15.4 ± 2.2 ^a^	587.3 ± 89.6 ^a^
*Ppck-8*	138.0 ± 5.1	12.5 ± 1.4	11.4 ± 2.3	9.4 ± 1.8	391.8 ± 98.4
*Ppck-9*	92.5 ± 7.4 ^a^	10.9 ± 0.7 ^a^	3.5 ± 0.9 ^a^	1.6 ± 0.7 ^a^	42.7 ± 18.0 ^a^
*Ppck-10*	135.4 ± 9.8	12.1 ± 1.1	8.0 ± 2.3	7.1 ± 2.0	309.8 ± 89.3
*Ppck-11*	126.6 ± 6.0 ^a^	12.5 ± 0.9	10.0 ± 1.4	8.2 ± 1.0	300.2 ± 47.5
*Ppck-12*	139.6 ± 5.6	14.6 ± 0.8	13.5 ± 1.4	11.0 ± 1.2	365.4 ± 64.7

Data were recorded at the time of harvesting after 5 months in a greenhouse. Data are means ± SE (*n* = 5) per panicle. ^a^ *p* < 0.05 versus Wt.

## Data Availability

All relevant data can be found within the manuscript.

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
