# Peer review of "Silencing of SbPPCK1-3 Negatively Affects Development, Stress Responses and Productivity in Sorghum"

_plants, 2023, doi:10.3390/plants12132426_

Round 1

Reviewer 1 Report

In the manuscript, Pérez-López et al. silenced three phosphoenolpyruvate carboxylase homologs in sorghum by RNAi and observed subsequent effects in two independent transgenic lines. Transgenic plants exhibited reduced expression of PPCK1, PPCK2, and PPCK3. PEP carboxylase binds CO2 in the form of bicarbonate with PEP to create oxaloacetate. This is then converted back to pyruvate (through a malate intermediate) to release the CO2 in the deeper layer of bundle sheath cells for carbon fixation by RuBisCO and the Calvin cycle. Thereby, the oxygenase property of rubisco at higher temperatures is compensated. That is the specialty of C4 physiology. The authors performed a lot of experiments. However, a clear and reasonable hypothesis to be tested is not presented. It is difficult to elucidate its function only by downregulating three among a total of five homologs. Surprisingly the authors did not consider testing the effects of heat. Although the shown effects of silencing the three can be a descriptive study, I do not recommend its publication without a major revision.

Why are only three genes targeted?

In abstract: The quality of seeds was lower in Ppck lines. Quality at what aspects? please specify

The authors discussed PTM and the phosphorylation of PEP carboxylase in the introduction and results section. However, I need to be more convinced about how a low level of mRNA production is associated with the regulation of phosphorylation.

Tables and graphs require more rigorous statistical tests and comparisons among all instead of wild-type only.

As claimed in the title, the tested stresses (salinity and low phosphate) do not represent stress to establish the role of PEP carboxylase in a system designed for heat-responsive physiology. Fatty and amino acid, C, and N content profiling also cannot claim its role in 'development.'

Figures are of low-quality for visibility. Fig 7b has no significance, use NCBI accession instead. I would love to see some picture in developmental alterations calimed.

Reviewer 2 Report

Dear Authors,  Bolow you can findmy opinion about manuscript ID  2433405. 

The manuscript titled "Silencing of SbPPCK1-3 negatively affects development, stressresponses and productivity in sorghum" is consistent with the aim of PLANTS and meets the formal requirements set by this journal. The authors of the manuscript touched on a current research problem. The abstract contains concise information. In the Introduction chapter, the authors introduce the reader to the research topic in great detail, describe the physiological, biochemical and genetic aspects of PEPC and the consequences of silencing PPCK genes and post-translation modifications of enzymes. Undoubtedly, this chapter is the strongest part of the presented work. Unfortunately, at the end of the introduction, I lack a clearly defined research goal and I did not find it in the whole work. In the Results chapter, the authors presented the results of all the analyses. They are legible. A brief description is provided for each result.

In the chapter "Material and methods", the authors described the research methods in detail, but they did not sufficiently describe the conditions of cultivation and treatment of plants. The "Discussion" chapter is by far the weakest part of the work. In my opinion it needs to be completed. The big downside is the lack of a "Summary".

Below is a list of questions and content to be completed in the manuscript.

1. The aim of the work / research hypothesis should be clearly stated.

2. What does the in vitro PEPCk antigen test mean?

3. Authors write that the use of the Pro-Q/SYPRO method was not sensitive enough to study PEPC activity. And this is my question: was the method too sensitive and what other method do the authors propose? or were there simply no differences in the activity of this enzyme?

4. Why were some analyzes performed only on one Ppck-2 line and the WT variant?

5. Stats definitely need to be improved and supplemented.

6. The expression of the PCK2 and PPCK3 genes was also observed in the roots, why was the activity of the enzymes not tested there as well? It is known that the root is not a photosynthetic organ, however metabolomic information from the roots is also transported to higher parts of the plant, where it can induce further cascades of biochemical transformations.

7. Figures 2 and 3 describe the treatment conditions of the plants with NaCl and Ca3(PO4)2(PCa). this information must necessarily be completed in the Material and Methods section.

8. The authors must explain why they analyzed the fatty acids. Here I would also like to ask the authors whether they determined 12 fatty acids, whether there were other acids, whether such patterns were only available, why were these acids determined? should a chromatogram from such a chapter also be included?

Ciewal for me is also a double increase in the content of Azelaic acid which induces the secretion of salicylic acid, and this in turn is one of the elements of plant resistance to stress.

9. The discussion should be comprehensively reviewed. It is laconic, in fact there is no explanation of what the authors studied.

10. I would also like to ask the authors for what purpose they performed the analysis of tannins in seeds? i.e. I can guess what the authors wanted to show, but this information is nowhere to be found in the manuscript.

11. The authors will include a lot of results in the work, therefore it is necessary to include a summary.

In conclusion, I believe that the work can be published in PLANTS, but only after taking into account the comments I have mentioned.

Best regards,

Reviewer

Round 2

Reviewer 1 Report

I thank authors for revising the manuscript. I have a few suggestion to improve it:

1.      Please re-write abstract for clarity. Also address these in the main text:

2.      Among them, two T2 homozygous lines (Ppck-2 and Ppck-4) were selected for further evaluation.

3.      Please revise “PPCK1 gene expression was….”  “ expression of SbPPCK1 ….. add Sb throughut the manuscript when you are specifically saying about this gene.

4.      Both leaves and seeds of Ppck lines had altered metabolic profiles, being remarkable a general decrease amino acids. Considering deleting being remarkable

5.      In addition, Ppck lines showed delayed flowering by how many day? and about 20% of Ppck-4 plants did not produce flowers. Not flower at al? Please clarify. How about Ppck-2?

6.      Total amount of seeds was lowered by 50% and 36% in Ppck-2 and Ppck-4 lines, respectively.

7.      Statistics: Performance difference of Ppck lines in various parameters should be compared wild-type. In these cases, showing the statistical difference is important to readers for easy comparison, even though they are statistically not significant. Please revise all tables and figures for this (by putting a, b, c etc). Please remove “P<0.05 versus Control” from the figure, mention in the legend instead.

Reviewer 2 Report

Dear Autors,

I am fully satisfied with the response of the authors to my comments. The authors also made the required changes to the paper. I recommend the publication of this manuscript in PLANTS. 

I look forward to reading your next report.

Kind regards, 

Reviewear

Author Response

The authors thank reviewer 2 the detailed revision of the paper to make it better